# The Proactive Effects of Built Environment on Rural Community Resilience: Evidence from China Family Panel Studies

**DOI:** 10.3390/ijerph20064913

**Published:** 2023-03-10

**Authors:** Xiaowan Dong, Yuhui Xu, Xiangmei Li

**Affiliations:** 1School of Architecture and Urban Planning, Chongqing University, Chongqing 400030, China; 2Cooperative Innovation Center for Emissions Trading System Co-Constructed by the Province and Ministry, Wuhan 430205, China; 3School of Low Carbon Economics, Hubei University of Economics, Wuhan 430205, China

**Keywords:** rural community resilience, objective built environment, perceived built environment, place attachment, China

## Abstract

Rural community resilience (RCR) is crucial to rural sustainable development in the context of rural decline globally. Previous studies seem to underestimate the role of the built environment (BE) in the proactive aspect of RCR (P-RCR), that is, a rural community’s ability to cope with change proactively. This study explores BE’s effects on P-RCR with a holistic framework involving objective BE (OBE), perceived BE (PBE), place attachment (PA) and P-RCR, using structural equation modeling (SEM) based on a sample of 7528 rural respondents from eastern, central and western China. The results are as follows: (1) Both OBE (population density and accessibility) and PBE (perceptions of facilities, surrounding environment and safety) can significantly affect P-RCR in terms of social, economic and environmental dimensions. (2) In all regions, PBE’s impacts were consistent and positive on social and economic dimensions at both the individual and community levels (except the community-level economic dimension in western regions), but negative on the individual-level environmental dimension; OBE’s impacts were varied among regions. (3) In certain regions, PA and PBE were mediators in the BE-P-RCR relationship. This study can help researchers to construct a more detailed picture of the BE-P-RCR relationship and identify BE-related factors that contribute to P-RCR enhancement.

## 1. Introduction

Fostering rural community resilience (RCR) is gaining increasing attention along with a series of rural issues confronted by many rural communities around the world, such as depopulation, economic depression, employment reduction and increasing disaster vulnerability [1,2,3,4]. RCR explains rural communities’ reactive and proactive responses to disturbances for their survival and development [5,6,7], providing new theoretical perspectives and strategies for rural communities to deal with the abovementioned issues [2,3,4,8]. Two aspects of RCR have been observed [6]. The reactive aspect of RCR (R-RCR), which ensures a rural community’s original state of maintenance and short-term recovery in the face of disturbances, typically involves the community’s ability to resist and absorb disturbance [6,9]. Additionally, the proactive aspect of RCR (P-RCR), which facilitates a rural community’s long-term survival and prosperity despite constant changes, usually combines personal and collective ability to respond to change proactively with diverse community resources [5,6,10]. Through this proactive aspect, resilient rural communities can deliberately utilize and develop those resources to adapt to change and transform themselves into a new state that is usually more resilient than the original one [6,10,11]. The enhancement of RCR not only improves the viability of rural communities exposed to fast-onset disasters, but also allows them to adapt to slow-onset demographic, socioeconomic and environmental changes more successfully and find new development pathways to overcome adversities [7,12,13]. Fostering RCR has been viewed as essential to reduce and prevent disaster risks in rural areas [3], to maintain rural populations, to improve rural life quality and diversify rural economies in the context of rural decline [2,8,14].

As a result, researchers have shown great interest in factors that enhance or undermine RCR [15], including built environment (BE) factors [16]. BE refers to “all of the physical structures and elements of the human-made environments in which we live, work, travel, and play” [17], as well as the design and planning of these structures and elements (e.g., urban design, land use planning, building codes) [18]. Whether a community can withstand and rapidly recover from disasters usually depends on the performance of BE during these disasters [19], and BE is also seen as an important resource for communities coping with constant change [10]. In previous studies, great attention has been paid to BE’s impacts on R-RCR. A range of BE attributes or components that dramatically influence the level of R-RCR have been identified, including lifelines and critical infrastructure, quality of building construction, land use planning, and design codes [16,20]. However, with respect to P-RCR, the effects of BE are either omitted [21] or limited in a facility-economy way. Most researchers tend to simplify BE to “infrastructure” or “facility” and view it as one of the indicators comprising the economic dimension of P-RCR due to its monetary value for rural communities [22,23,24]. For example, facilities provide services for the needs of people and companies, attracting businesses that help rural communities develop their economic resources [25]. One of the reasons leading to RCR researchers’ disproportional attention to BE might be that the role of BE in P-RCR is not as explicit as in the process of rural communities withstanding disasters.

Nevertheless, studies in rural health, psychology and environmental psychology indicate that BE, either objectively measured (OBE) or perceived (PBE), is not dispensable to P-RCR and has multiple approaches to influence P-RCR not merely through the facility-economy way. Researchers focusing on rural health and the resilience of individuals find that “facilities” also are of non-monetary value to rural communities and their resilience by offering locations for people’s social interactions and influencing social networks through perceived availability (PBE attribute) [26,27,28]. The non-facility BE attributes, for example, perceived aesthetics of buildings and streets (PBE attribute), have impacts on rural communities’ economic diversity by attracting settlers to the area [28], and objectively measured population density (OBE attribute) is associated with the environmental conditions of rural communities [29]. Those social, economic and environmental factors are the key elements that constitute three fundamental dimensions of P-RCR [30]. Moreover, environmental psychology researchers note that place attachment (PA) can be impacted by OBE or PBE [31], while PA is an important factor closely related to P-RCR [32]. These findings imply that OBE/PBE may indirectly affect P-RCR through PA.

However, although a few researchers test the links between rural people’s perceptions of facilities and P-RCR in case studies [2], the implications of OBE and PBE have not yet been further examined simultaneously and holistically in empirical studies on P-RCR. In addition, there is also a lack of a framework that depicts BE’s effects on P-RCR from an integrative perspective. To fill this gap, we explored whether and how OBE and PBE affect P-RCR in different dimensions with a holistic framework and structural equation modeling (SEM) based on a sample of 7528 rural community residents from China. This sample was nationally representative and divided into eastern, central, and western region groups according to communities’ geographic locations. The reasons we selected Chinese rural communities as our research objects were as follows. First, compared to developed countries, resilience research on communities attracts limited attention in developing economies, with a particular gap in research in China regarding P-RCR [33]. Moreover, these communities are experiencing demographic and socioeconomic changes that have been seen as global issues in rural areas or termed as rural decline [1]. Therefore, research on the relationship between BE and P-RCR in China will help domestic as well as international researchers develop a more detailed picture of RCR in the context of rural decline globally. Second, what those resilience communities need to improve urgently is the proactive aspect. Along with rapid urbanization and industrialization, the rural population in China began to decrease dramatically in 1995 [34]. The size of the rural population declined by about 0.36 billion from 1995 (0.86 billion) to 2021 (0.50 billion) [35]. The outmigration of rural laborers has accompanied this, leading to the reduction in human resources in traditional agriculture, a decline in the agricultural income of rural households, hollowed-out villages and the deterioration of traditional values [36]. To solve rural problems including depopulation, lack of economic opportunity and the weakening of agricultural social cohesion, a “rural revitalization strategy” was proposed in China in 2017 [37]. As a continuation of this strategy, the Chinese government issued the “Rural Revitalization Strategic Plan (2018–2022)” in 2018, with the revitalization of rural communities as its essential part [38]. Against this backdrop, fostering P-RCR in China is critical and urgent because, for revitalization and sustainable development, rural communities undergoing these changes require intentional adaptation and transformation rather than maintenance of their original state, which hardly exists amid constant socioeconomic change [13]. Further, those changes also weaken RCR in China, including P-RCR [33,38]. In the face of such changes, the resilience of rural communities faces challenges in community resources as well as residents’ willingness and capacity to assume responsibility for community development [39]. Third, exploring the BE-P-RCR relationship is key to P-RCR enhancement and sustainable rural reconstruction in China. Top-down and bottom-up BE reconstruction have long been seen as important approaches to reverse the trend of rural decline and as strategies for achieving rural renaissance in China [40,41]. A series of policies with a central focus on BE have been implemented since 2005, including new rural reconstruction and scenic rural development [42]. However, though remarkable achievements have been made in these reconstruction efforts, there are criticisms that farmers’ interests and social connections are often ignored in those BE transformations, causing social contradictions [41,42] that decrease P-RCR. As a result, investigating BE’s influences on P-RCR is necessary for a better rural BE and improved P-RCR in China. It can be useful for planners and architects identifying specific BE attributes that reinforce P-RCR as well as contribute to sustainable rural reconstruction.

Specifically, in this study, we focused on the following questions: Do OBE and PBE significantly affect P-RCR in the social, economic and environmental dimensions?How do OBE and PBE affect these dimensions, respectively?Does PA or PBE play a mediating role in the BE-P-RCR relationship?

## 2. Theoretical Framework of the BE-P-RCR Relationship 

We propose a holistic framework (Figure 1) to depict the BE-P-RCR relationship based on an extensive literature review on BE, P-RCR and PA, which we will explain in the following subsections. We applied this framework as our research scheme and the basis of the structural equation models we used for statistical analysis in Section 3. In this framework, there were six paths in total, including the paths from OBE to PBE, OBE to P-RCR, OBE to PA, PBE to P-RCR, PBE to PA and PA to P-RCR. These paths represent the OBE-PBE relationship and the potential ways OBE and PBE affect P-RCR in the social (Soc), economic (Eco) and environmental (Env) dimensions. 

### 2.1. The Path from OBE to PBE

We used both PBE and OBE to explore BE’s impacts on P-RCR, and we recognized that OBE has effects on PBE based on the framework proposed by Marans [43]. Except for distinctions in measurement approach, the differences between OBE and PBE have led to considerations of the OBE-PBE relationship [43,44] and the issues of relying solely on one approach to explore BE [45]. Many researchers have noticed that OBE cannot be equal to PBE, even considering the similar environmental attributes, since different people might have different views on the same objective attributes [43,44]. To further explain this, Marans [43] asserts that PBE reflects people’s perceptions and assessments of OBE, which is influenced by their past experiences and OBE itself, and OBE has impacts on people’s satisfaction with their community through PBE. In addition, Lewicka [45] suggests that depending merely on residents’ perceptions of BE in PA studies is less reliable due to the biases that may exist in these perceptions. As a result, our framework incorporated both OBE and PBE with a pathway from OBE to PBE.

### 2.2. Direct Paths from OBE/PBE to P-RCR with Three Fundamental Dimensions

#### 2.2.1. Three Fundamental Dimensions of P-RCR

Social, economic and environmental capital are fundamental to P-RCR. P-RCR relies on the personal and collective agency of members [6,11,46], as well as community resources or capital (e.g., social, economic, environmental, and cultural capital) that can be deployed to deal with change [10,11,47]. It is enhanced through deliberate development and engagement of these resources or capital by community members [10], and well-developed capital often represents a high level overall or specific dimensions of P-RCR [22,23]. Soc, Eco and Env are seen as fundamental dimensions of P-RCR [5], because they embody community members’ willingness and ability to work together [10] as well as the critical resources or capital communities use to respond to change [11,47]. Soc, including factors such as the social networks between individuals and groups [30], trust [30], and happiness [13], is fundamentally about the community members’ willingness and capacity to participate in actions for coping with change [10]. Eco is the financial base of a rural community and its members and includes components such as community economic well-being [47], individual financial stability [13], and economic diversity [24]. Env often refers to the ecological conditions of a rural community such as soil conditions [47], water quality [47] and biodiversity [48], and the pro-environmental attitudes or behaviors of the community members [5].

#### 2.2.2. The Influences of OBE/PBE on P-RCR

The existing work suggests that OBE/PBE may influence P-RCR in terms of all three dimensions. Regarding Soc, BE components such as schools, stores, and recreational and healthcare facilities provide physical spaces for rural residents’ social interactions [26,28]. Consequently, the low perceived availability of facilities or infrastructures weakens rural people’s social networks and lead to a decrease in resilience [28]. Moreover, although lacking validation in rural communities, researchers find that objectively measured accessibility and perceived adequacy of facilities have independent impacts on social capital in suburban communities [49], and perceptions of safety are associated with social capital in urban communities [50]. Concerning Eco, facilities contribute to the rural economy in multiple ways (e.g., financial and retail services, job provision, tourism encouragement, consumption) [27], which implies that accessibility may be important for the economic dimension of P-RCR. Meanwhile, the attractiveness of BE, such as aesthetic perceptions of buildings and streetscapes, is helpful in the economic diversification of rural communities [28]. Regarding Env, it is plausible that some PBE attributes (e.g., perceptions of litter and refuse) are associated with garbage pollution in rural China [29]. Some researchers also notice that insufficient facilities decrease residents’ willingness to participate in environmental projects [51], and the population density (objectively measured) of Chinese rural communities can influence their environment [29]. 

#### 2.2.3. Direct Paths from OBE/PBE to Different Dimensions of P-RCR

It should be noted that the abovementioned influences of OBE/PBE on P-RCR might be direct, indirect or both. We assume that OBE/PBE affects P-RCR in both ways. Direct paths from OBE/PBE to P-RCR are included in the framework.

### 2.3. The Path from PA to P-RCR

PA is described as “an emotional connection to a place” [52] (p. 560). Usually, PA has been seen as a good thing for P-RCR, since it motivates rural people’s participation in community organizations and helps them cope with social, economic and environmental problems and disasters in most cases [32,53], though some researchers also notice the adverse effects of certain kinds of PA on RCR [54].

### 2.4. The Path from OBE/PBE to PA

In rural or agriculture studies relevant to BE, researchers have found the importance of OBE/PBE to PA. For example, Bunkus et al. [55] emphasize that population density reflecting the quantity of interactions impacts farmers’ PA in Germany directly and indirectly; Christiaanse and Haartsen [56] confirm that the decreasing numbers of rural facilities have disrupted the PA between rural people and these facilities and resulted in negative emotional reactions and collective actions; researchers also recognize that within the context of Chinese rural land consolidation, rural residents’ perceptions of BE are closely related to their place identity [57], which has been viewed as an important component of PA [58]. 

## 3. Materials and Methods

### 3.1. Data

The data used in this study were derived from China Family Panel Studies (CFPS). CFPS, conducted by the Institute of Social Science Survey (ISSS) of Peking University, is a national and comprehensive social survey aiming to collect longitudinal data at the individual, family and community levels in contemporary China for research on Chinese social phenomena [59]. It covers 25 provinces, municipalities or autonomous regions in China (except Hong Kong, Macao, Taiwan, Xinjiang, Qinghai, Inner Mongolia, Ningxia and Hainan) and is carried out in waves every 2 years [59]. The data of CFPS contain many datasets, including datasets related to communities and adult family members. We used different datasets and waves of CFPS (Table 1), because the variables in our study involved many aspects of rural life that connect with several CFPS datasets released so far, and parts of these variables were collected in different waves. 

Specifically, we combined the variables of OBE and P-RCR (Soc, Env and part of Eco) from CFPS 2014 (datasets on communities and adults), the variables of PBE, P-RCR (Eco pertaining to individuals) and PA from 2016 (adult dataset) by linking “community ID” after keeping all samples of rural communities (communities in rural areas and urban residential areas newly transformed from villages). However, we excluded the respondents who had moved to a new residential address or had a primary job and income change during 2014–2016, for these respondents may make less reliable evaluations of OBE and individual economic well-being in the context of our study. At last, we obtained a sample of 7528 rural community respondents in China. 

We divided this sample into three groups based on the geographic locations of 25 provinces (municipalities or autonomous regions). These were the groups of eastern regions (n = 2719), central regions (n = 2130) and western regions (n = 2679) (Table 2). This was because besides the community-scale factors, factors outside the community can also influence RCR (e.g., regional policies, markets and natural resources) [13,22]. External factors, such as unbalanced rural industrial development and rural income inequality in coastal and inland regions in China [60,61], could interfere with our study, which concentrated on community-scale BE impacts on P-RCR. As a result, the effects of BE on P-RCR in this study were explored separately using three groups of data. STATA Version15 was used for data cleaning and grouping. 

### 3.2. Variables

#### 3.2.1. OBE

For our research purpose, OBE in this study was quantified with accessibility and population density [49,62,63]. We measured these two variables using the equations proposed by Sun et al. [64]. In CFPS 2014, the data relating to OBE included size of community administrative area (square kilometers), permanent resident population and numbers of facilities (stores, kindergartens, primary schools, middle schools, hospitals or clinics, pharmacies, churches, ancestral halls, temples, activity facilities or community service centers for the elderly, nursing homes, physical exercise facilities and playgrounds) in the community. Based on Sun et al.’s study [64], we treated population density and accessibility of facilities as two observation variables of OBE and calculated them as follows:d = P/A(1)
a = N/A(2)
where d is population density, a is accessibility, P is permanent resident population of the community, N is the number of facilities in the community, A is size of administrative area (square kilometers). A higher proportion accounted for higher population density and better accessibility.

#### 3.2.2. PBE and PA

For PBE assessment, we used the data on residents’ perceptions of their neighborhood BE in CFPS 2016. It contains overall perceptions of public facilities, safety and surrounding environment of neighborhood (e.g., aesthetics and noise), which have been seen as critical PBE attributes in studies pertaining to a rural context, social capital or PA [65,66]. We treated PBE as a latent variable consisting of these three kinds of perceptions. PA was estimated using data from the residents’ evaluations of their emotional attachment to the community in CFPS 2016. All indicators of PBE and PA were rated with a 5-point scale, ranging from 1 (very good or very much) to 5 (very poor or not at all). We reversed the code for the convenience of explaining that a higher score represented a better perception of BE.

#### 3.2.3. Key Dimensions of P-RCR

In this study, P-RCR was quantified based on the framework explored by Markantoni et al. [5], which integrated the frameworks proposed by Steiner and Markantoni [13] and Wilson [30] to measure Soc, Eco and Env at the individual and community levels. Since this framework focuses on socioeconomic changes of rural areas and estimates the three key dimensions of P-RCR through both individual and collective levels, it seemed appropriate for our study.

At the individual level, Soc was assessed by happiness [13], and based on Shen and Jia [67], we used self-evaluated happiness (10-point scale, ranging from “lowest” to “highest”), life satisfaction (5-point scale, ranging from “very unsatisfied” to “very satisfied”) and confidence in the future (5-point scale, ranging from “not confident at all” to “very confident”) derived from CFPS 2014 as the measurement indicators of happiness. Individual-level Eco was evaluated by personal financial stability [13], and the data on income satisfaction (5-point scale, ranging from “very unsatisfied” to “very satisfied”), overall job satisfaction (5-point scale, ranging from “very unsatisfied” to “very satisfied”) and working environment satisfaction (5-point scale, ranging from “very unsatisfied” to “very satisfied”) obtained from CFPS 2016 were used to measure this stability. Individual-level Env was estimated on the basis of pro-environmental attitudes or behavior [5], using the severity of environmental problems rated by adult respondents (10-point scale, ranging from “not severe” to “extremely severe”) in CFPS 2014 as the indicator. This was because people facing severe environmental problems are more likely to support environmental protection [68]. At the community level, Soc, Eco and Env were measured in terms of trust in neighborhood [30], community economic well-being [47] and biodiversity [48], respectively. The data obtained from CFPS 2014 (adult dataset) were used to quantify these dimensions, including neighborhood trust (10-point scale, ranging from “distrustful” to “very trustworthy”), net income per capita (CNY) and the proportion of forest and/or land with fruit trees in the community administrative area. For all levels, a higher rating score, income or proportion represented better PBE or greater Soc, Eco and Env. 

#### 3.2.4. Covariate

The covariate was the socioeconomic status of residents. Lewicka [45] asserts that the predictors of PA include physical factors (e.g., objective BE features) as well as social factors (e.g., safety, social ties), and the relative importance of these factors depends on residents’ socioeconomic status in some cases. Since PA was an important endogenous variable in our study, overlooking the differences in residents’ socioeconomic status might have led to imprecision in our study. Therefore, we used socioeconomic status as the covariate and employed the data on self-reported social and economic status (5-point scale, ranging from lowest to highest) obtained from CFPS 2014 (adult dataset) to measure this covariate.

#### 3.2.5. Questions Used for Variable Measurement

Specific questions used to derive indicators of PBE, PA, different dimensions of P-RCR and the covariate are displayed in Table 3.

### 3.3. Methods

Structural equation modeling (SEM) has been an important tool for analyzing the interactions between the physical environment and rural society [55,57]. SEM consists of a measurement model that can measure the reliability and validity of latent variables, and a structural equation that can be used to analyze the paths between the constructs. 

The reasons we employed SEM as an analytical tool in this study were manifold. First, SEM allows researchers to investigate complex relationships between multiple constructs in a single model and provides an easier way to discuss the model [55,69]. Therefore, it fit well with our study, which attempted to explore the associations between OBE, PBE, PA and three different dimensions of P-RCR in one holistic framework. Second, SEM is usually applied to verify a theoretical hypothesis by analyzing observations and latent variables through statistical procedures including path analysis, regression and structural equations [55,57]. As a result, it could be a useful tool for testing the BE-P-RCR relationship we postulated in this study. The statistical analysis in this study was built on three steps: 1.Step one: Measurement model testing and descriptive statistical analysis

In this step, confirmatory factor analysis (CFA) was performed using different groups of data and AMOS Version 26. Meanwhile, descriptive statistical analysis was conducted using STATA Version15.

2.Step two: Structural equation model building

In this step, two basic structural equation models were established based on the framework shown in Figure 1, including the individual-level model (with the variables of OBE, PBE, PA and Soc, Eco, Env at the individual level) and the community-level model (with the variables of OBE, PBE, PA and Soc, Eco, Env at the community level). Figure 2 demonstrates the structure of the two basic models. 

3.Step three: Application of structural equation model

Since population density and accessibility are highly correlated, these two indicators were examined separately in the basic models for multicollinearity reduction. Consequently, there were four models we needed to explore: Individual-level model with population density (Model 1);Individual-level model with accessibility (Model 2);Community-level model with population density (Model 3);Community-level model with accessibility (Model 4).

Each model was tested using three groups of data separately; therefore, 12 models were applied using AMOS Version 26, and the paths to all endogenous variables were controlled for the covariate.

## 4. Results

### 4.1. The Results of CFA and Descriptive Statistics 

All composite reliability (CR) values for latent variables with multiple indicators derived from three groups of data were above 0.6 (0.776 ≥ CR ≥ 0.631), indicating a high degree of internal consistency [70] (Table A1). For acceptable convergent validity, generally an average variance extracted (AVE) value should be 0.5 or above [70]. However, Chin [71] suggests that most loadings should be at least 0.60 to ensure each measure can explain half or more of the variance in the latent variable, which indicates that the threshold of the AVE value should be at least 0.36. In this study, all AVE values exceeded or were close to 0.5 (0.539 ≥ AVE ≥ 0.428), except AVE for PBE and individual-level Soc in western regions (above 0.36) (Table A1). Moreover, the square root of the AVE value for each latent variable with multiple indicators was greater than the values of its correlations with other multiple indicator variables, demonstrating a high discriminant validity of our models (Table A2). 

Table 4 displays the average population density (natural logarithm) and the standard deviation (sd) in eastern, central and western regions, which were 5.649 (sd = 1.461), 5.885 (sd = 1.352), and 5.196 (sd = 1.234), respectively. The average accessibility (natural logarithm) and the standard deviation in eastern, central and western regions were 0.728 (sd = 1.426), 0.855 (sd = 1.353), and 0.239 (sd = 1.263), respectively. Most respondents evaluated PBE as fair, as the median values for public facilities, surrounding environment and public safety in three regions were all 3 (“fair” option), while the interquartile ranges (IQR) of these values were all 1, which represents a central tendency of these values. Moreover, respondents are somewhat emotionally attached to their communities (all Median = 4, IQR ≤ 2). On average, respondents reported similar levels of life satisfaction (Mean ≈ 3.8) and confidence in their future (Mean ≈ 4.1) in all regions, but higher happiness levels in the eastern and central regions (Mean ≈ 7.5) than in the western regions (Mean ≈ 6.9). Most respondents reported their income (all Median = 3, IQR ≤ 2), working environment (all Median = 3, IQR = 1 except Median = 4 in central regions) and overall job satisfaction (all Median = 3, IQR = 1) as fair. Eastern and central region respondents reported higher average severity of environmental problems, neighborhood trust degree and net income per capita (natural logarithm) than their western region counterparts. The average biodiversity (natural logarithm) and the standard deviation were 6.978 (sd = 5.765) in eastern, 4.453 (sd = 5.531) in central and 6.925 (sd = 5.853) in western regions.

### 4.2. Analysis of the Results of the Structural Equation Model

All 12 models had acceptable goodness of fit. Because in each model, the Chi-square/degrees of freedom < 5, the comparative fit index > 0.95, the root mean square error of approximation < 0.05, and the standardized root mean square residual was below 0.05. The specific fits of each model are shown in Appendix A Table A3. As shown in Figure 2, direct effects included the impacts of direct paths from OBE/PBE to three dimensions of P-RCR. The indirect effects consisted of the impacts of paths from OBE to three dimensions of P-RCR through PA and first PA, then PBE, as well as the paths from PBE to three dimensions of P-RCR via PA. Total effects were the sum of direct effects and indirect effects. It represents all of the effects an exogenous variable had on an endogenous variable in this study. 

To examine whether and how BE influenced three dimensions of P-RCR, we first focused on whether there were positive or negative significant total effects of OBE/PBE on three dimensions of P-RCR, since it was more rational to infer that BE can affect P-RCR when significant total effects of BE are identified. Then, we paid attention to the indirect effects that PA or PBE can mediate. We highlighted the mediating role that PA and PBE play in the BE-P-RCR relationship when the total effects of OBE/PBE on P-RCR are significant. This study did not elaborate on the significant mediation effects related to insignificant total effects.

#### 4.2.1. Total and Indirect Effects of PBE on P-RCR

Table 5 shows that the total effects of PBE on Soc and Eco were significant and positive in all three regions at the individual and community levels, apart from community-level Eco in the western regions (significant and negative). Regarding Env, significant and negative total effects of PBE were identified at the individual level in all regions, while at the community level, the total effects of PBE were negative in the eastern regions, positive in central regions and insignificant in western regions. 

In eastern regions, the mediation effects of PA were found in the relationships between PBE and Soc/Eco at the individual and community levels. In central regions, PA significantly mediated the effects of PBE on Soc (individual- and community-level) and individual-level Env. No mediation effect of PA was observed in the western regions.

#### 4.2.2. Total and Indirect Effects of OBE on P-RCR

The values in Table 6 illustrate that the total effects of population density and accessibility on Soc were significant and negative in the eastern regions at the individual level. Referring to Eco, in the eastern regions, a significant and negative total effect of accessibility on Eco was only identified at the individual level. The values listed in Table 7 and Table 8 show that in the central and western regions, both accessibility and population density had significant total effects on Eco at the individual and community levels. Regarding Env, the total effects of population density and accessibility were significant at the community level but insignificant at the individual level in all regions. 

Concerning indirect effects, the values in Table 6 indicate that both PA and PBE significantly mediated the influences of OBE (population density/accessibility) on individual-level Soc and accessibility on individual-level Eco in the eastern regions. However, when considering all of the indirect paths from OBE to individual-level Soc in the eastern regions, OBE had insignificant total indirect effects on individual-level Soc, since the effects of OBE on Soc through the PBE path (negative) and firstly PA, then the PBE, path (negative) offset the PA path effects (positive). The values in Table 6, Table 7 and Table 8 demonstrate that PBE could also be a mediator in the relationship between OBE and community-level Env in the eastern and central regions, as well as between OBE and Eco in the western regions.

## 5. Discussion

### 5.1. Significant Effects of PBE/OBE on Three Dimensions of P-RCR

Our findings show that both PBE and OBE significantly affected three dimensions of P-RCR. PBE had significant total effects on three dimensions of P-RCR in all regions at both the individual and community levels (except community-level Env in western regions). OBE significantly affected individual-level Soc (eastern regions), Eco (individual- and community-level Eco in central and western regions; individual-level Eco in eastern regions) and community-level Env (all regions). These findings statistically support that the way BE influences P-RCR is varied rather than constrained in a facility-economy approach.

### 5.2. Differences between Effects of PBE/OBE on Three Dimensions of P-RCR 

PBE’s impacts on P-RCR were consistent among regions regarding Soc, Eco and individual-level Env. PBE was positively associated with Soc and Eco in all regions at both levels, with the single exception of community-level Eco in the western regions. This indicates that better PBE leads to a greater P-RCR in Soc and Eco in most cases. This is in line with the findings based on a Western context that perceived adequacy of facilities, attractive BE and feelings of safety contribute to richer social capital and greater economic resilience [28,49,50]. Referring to Env, a negative link between PBE and individual-level Env was observed. One of the potential explanations for why PBE predicts residents’ negative attitudes toward environmental protection is that residents’ willingness and actions to protect the environment are frequently associated with environmental deterioration [68]. However, good PBE is more likely to correlate with a good residential environment. In terms of community-level Env, PBE’s impacts showed an inconsistency in different regions. In central regions, better evaluations of PBE increased the likelihood of higher forest (and/or fruit tree) coverage in rural communities. It may be evidence of the finding that sufficient facilities or infrastructure raise farmers’ willingness to participate in the Grain-for-Green Project in China [51]. However, this was not applicable for explaining the correlations between PBE and community-level Env in the eastern (negative) and western regions (insignificant). Therefore, further research is needed to ascertain the relationship between PBE and community-level Env.

Compared to PBE, OBE’s impacts on the three dimensions of P-RCR were varied in different regions and levels. This implies that regional disparities may be critical to OBE’s impacts on P-RCR, and OBE plays different roles in fostering individual- and community-level P-RCR. For instance, our results show that in the eastern coastal regions where rural industrial development levels were higher and rural income inequality levels were lower [60,61], higher population density or accessibility may result in lower levels of happiness and undermine Soc, but no similar correlation was found in inland central and western regions where rural industrial development levels were lower, and income inequality levels were higher. In terms of OBE’s impacts on different levels of P-RCR, in central regions, higher population density or accessibility may have contributed to a greater Eco at the individual level but not at the community level; the exact reverse was the case in the western regions. 

### 5.3. The Significant Mediation Roles PA and PBE Play in the BE-P-RCR Relationship

Our findings indicate that there are two significant mediation roles that PA plays in the BE-P-RCR relationship. One is the positive role in P-RCR enhancement. PA provides a critical indirect path through which better PBE bring an increment in Soc (eastern and central regions) and individual-level Eco (eastern regions). It counteracts the negative effects of OBE (accessibility/density) on happiness and OBE (accessibility) on personal financial stability in the eastern regions. It offsets the adverse effects of PBE on residents’ attitudes toward environmental protection in the central regions. The other is the negative role in P-RCR enhancement. In the eastern regions, it reduces the positive impacts PBE has on community economic well-being. This suggests that the mediation effects of PA are not always beneficial to P-RCR, which is in line with the findings that different types of PA play different roles in RCR [54]. For example, residents with stability-oriented PA are often unwilling to change their current economic lifestyle closely related to rural facilities and services, which may not be good for enhancing community-level resilience [54]. 

In the western regions, OBE significantly influences Eco through PBE, and in eastern regions, accessibility has significant impacts on individual-level Eco through PBE. This supports the OBE-PBE relationship proposed by Marans [43] and the impacts of OBE/PBE on Eco. PBE also mediates the influences of OBE on community-level Env in the eastern and central regions, although the mechanisms remain unclear.

### 5.4. Strengths and Limitations 

This study focused on the questions of whether and how BE affects P-RCR, subjects that attracted less attention in previous studies but are crucial to RCR enhancement. We investigated the effects of OBE/PBE on three dimensions of P-RCR using SEM based on the empirical data obtained from CFPS and a holistic framework combining OBE/PBE, PA and P-RCR. Our findings provide a more-detailed picture of the BE-P-RCR relationship and new empirical evidence of BE’s multi-effects on P-RCR in terms of Soc, Eco and Env. 

This study has several limitations. First, it was difficult to determine whether BE positively or negatively influences the overall level of P-RCR based on our study, since OBE and PBE had inverse effects on the same dimension at different levels in specific regions, and their effects on different dimensions were also inverse sometimes. Nevertheless, our findings are useful for researchers in understanding BE’s impacts on specific dimensions and levels of P-RCR. Second, as stated earlier, the mechanism leading to PBE’s impacts on community-level Env is still unclear. This may be a result of our measurement strategies (e.g., using biodiversity for community-level Env measurement) and data limitation. Therefore, more data and sophisticated research designs are required to understand the relationships between PBE and Env in future study. Third, the data we used in this study was collected in different waves, which might not have been beneficial for our research in terms of reliability. However, many researchers recognize that it is acceptable to apply data obtained from different waves of CFPS in the same analysis or SEM model considering the reality of China [64,72]. Moreover, we restricted respondents’ residential addresses, income and professions when using 2014 and 2016 data for reliability improvement. As a result, applying data derived from 2014 to 2016 in this study would still be acceptable and reliable. Fourth, although our sample covered a large number of rural communities in China, our study was cross-sectional and limited to causality assessment. We could not verify the potential reciprocal causation between BE and P-RCR in this study, though resilient rural communities may intentionally increase built capital investments that improve BE qualities. This is not only because the exact pathways and scales by which P-RCR affects BE are still unclear, but also because the cross-sectional data can not statistically estimate reciprocal causation due to the lack of temporal precedence [73]. An improved model and panel data are needed in future studies. At last, the datasets we used were not recent, though they were the most recent datasets available in relation to BE in the released CFPS datasets. However, we believe the advantages of using these datasets were evident and that our findings are still valid for the current realities. The reasons are as follows. First, these datasets were longitudinal and nationally representative, providing high-quality and large sample-size data. Moreover, they can be updated in the future and facilitate our follow-up study with panel data. Second, at the core of our study was the BE-P-RCR relationship, which closely relates to psychological factors not easily changing with time, such as human cognition, emotions and behaviors. This means that for this study, the time factor might not be decisive. The consistency between our findings and some earlier study results discussed in the discussion section might be seen as evidence.

### 5.5. Implications

The results of our study imply that BE’s impacts on P-RCR are multifaceted and should be fully considered in RCR study and practice, especially with regard to Soc and Eco. When assessing or analyzing P-RCR in terms of Soc and Eco, rather than simplifying BE as facilities that purely increase rural communities’ economic resilience, specific PBE and OBE attributes should be taken into account. To enhance P-RCR, besides the number of facilities and population density, PBE may be a key factor. Furthermore, one-size-fits-all criteria for accessibility might not be appropriate in different rural communities, for the effects of accessibility on P-RCR may be uneven among regions. Additionally, greater consideration needs to be given to the influences of PA on Soc and Eco at both the individual and community levels (e.g., happiness, trust, satisfaction with job and income) when any BE changes occur within rural reconstruction.

Based on our findings, the following recommendations are offered for rural community development and P-RCR enhancement in China or other countries or regions facing rural issues similar to those in China:Improvements to the rural built environment, such as new rural reconstruction and rural settlement remediation, should not focus only on infrastructure development while ignoring people’s perceptions and evaluations of their surrounding environment.Top-down planning activities initiated by the government should develop more detailed and targeted planning schemes for rural service accessibility and village mergers, which will be helpful for increasing P-RCR in different regions.The development and implementation of built environment policies should consider rural people’s emotional ties with their communities, including both the pros and cons of these emotional ties for P-RCR.

## 6. Conclusions

To explore BE’s effects on P-RCR, this study proposes a framework holistically depicting the BE-P-RCR relationship and tested this framework using SEM with a sample of 7528 rural respondents from eastern, central and western China. Our findings include the following: (1) Both OBE and PBE can significantly affect three fundamental dimensions of P-RCR; (2) apart from community-level Eco in western regions, PBE consistently and positively influenced Soc and Eco but negatively influenced individual-level Env despite regional disparities; OBE’s impacts on three dimensions of P-RCR were varied among regions; (3) PA and PBE were mediators in the BE-P-RCR relationship in certain regions. Based on these findings, we argue that the multi-effects of BE on P-RCR should be taken into account in RCR research and practice, especially regarding Soc and Eco. 

## Figures and Tables

**Figure 1 ijerph-20-04913-f001:**
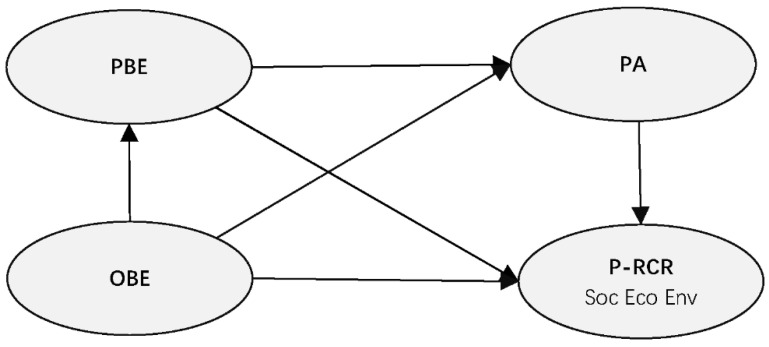
Theoretical framework of the BE-P-RCR relationship. PBE = perceived built environment; OBE = objective built environment; PA = place attachment; P-RCR = the proactive aspect of rural community resilience; Soc = social dimension of P-RCR; Eco = economic dimension of P-RCR; Env = environmental dimension of P-RCR.

**Figure 2 ijerph-20-04913-f002:**
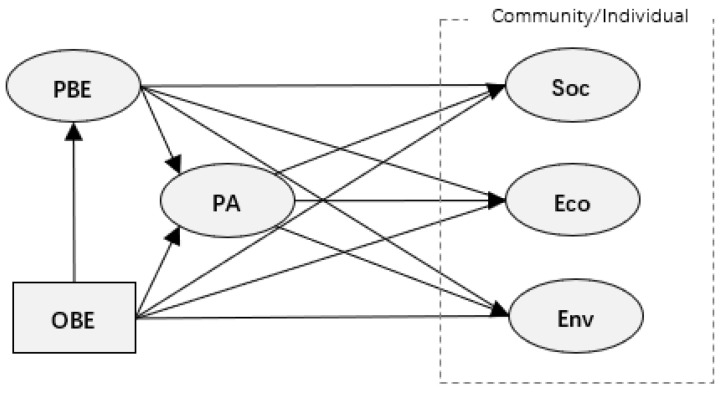
The structure of individual- and community-level models (Source: authors).

**Table 1 ijerph-20-04913-t001:** Datasets and CFPS waves used in this study.

Variables	Data Source (Datasets)	Data Source (Waves)
OBE	community	CFPS 2014
PBE	adult	CFPS 2016
PA	adult	CFPS 2016
P-RCR	community; adult	CFPS 2014
P-RCR	adult	CFPS 2016

OBE = objective built environment; PBE = perceived built environment; P-RCR = the proactive aspect of rural community resilience; CFPS 2014/2016 = China Family Panel Studies in 2014/2016.

**Table 2 ijerph-20-04913-t002:** The eastern, central and western regions.

Regions	Provinces, Municipalities or Autonomous Regions	Sample Size
Eastern	Beijing, Tianjin, Hebei, Liaoning, Shanghai, Jiangsu, Zhejiang, Fujian, Shandong, Guangdong, Guangxi	2719
Central	Shanxi, Jilin, Heilongjiang, Anhui, Jiangxi, Henan, Hubei, Hunan	2130
Western	Chongqing, Sichuan, Guizhou, Yunnan, Shaanxi, Gansu	2679

**Table 3 ijerph-20-04913-t003:** Questions used to derive indicators of PBE, PA, P-RCR and the covariate.

Variables		Indicators	Questions	Source
PBE		Community Environment	How is the surrounding environment of your community (noise, trash disposal, etc.)? (reversed code ranging from 1 = very poor to 5 = very good)	CFPS 2016 Full Questionnaires
		Safety	How is the public safety around your community? (reversed code ranging from 1 = very poor to 5 = very good)	
		Public Facilities	What do you think of the public facilities around your community? (reversed code ranging from 1 = very poor to 5 = very good)	
P-RCR	Individual-level	Social Dimension	1. Are you happy? (ranging from 1 = lowest to 10 = highest) 2. How confident are you about your future? (ranging from 1 = not confident at all to 5 = very confident) 3. Are you satisfied with your life? (ranging from 1 = very unsatisfied to 5 = very satisfied)	CFPS 2014 Full Questionnaires; CFPS 2016 Full Questionnaires
		Economic Dimension	1. How satisfied are you with your current income from this job? (ranging from 1 = very unsatisfied to 5 = very satisfied) 2. In general, how satisfied are you with this job? (ranging from 1 = very unsatisfied to 5 = very satisfied) 3. How satisfied are you with the working environment in this job? (ranging from 1 = very unsatisfied to 5 = very satisfied)	
		Environmental Dimension	How would you rate the severity of the environmental problem in China? (ranging from 1 = not severe to 10 = extremely severe)	
	Community-level	Social Dimension	How much do you trust your neighborhood? (ranging from 1 = distrustful to 10 = very trustworthy)	CFPS 2014 Full Questionnaires
		Economic Dimension	The net income per capita in your village (yuan)	
		Environmental Dimension	1.The total area of forest and/or land with fruit trees in your village (mu) 2. What is the current administrative area of your village/residential community? (kilometer^2^/mu)	
PA		Emotional Attachment	How would you rate your emotional attachment to your community? (ranging from 1 = very good to 5 = very poor)	CFPS 2016 Full Questionnaires
Covariate		Self-reported Socioeconomic Status	1. What is your relative income level in your local area? (ranging from 1 = lowest to 5 = highest) 2. What is your social status in your local area? (ranging from 1 = lowest to 5 = highest)	CFPS 2014 Full Questionnaires

BE = built environment; PBE = perceived built environment; P-RCR = the proactive aspect of rural community resilience; PA = place attachment. CFPS 2014/2016 = China Family Panel Studies in 2014/2016.

**Table 4 ijerph-20-04913-t004:** Descriptive statistics for the covariate, individual- and community-level variables derived from samples of the eastern (n = 2719), central (n = 2130) and western regions (n = 2679).

	Eastern Regions				Central Regions				Western Regions			
	(1)	(2)	(3)	(4)	(5)	(6)	(7)	(8)	(9)	(10)	(11)	(12)
Variables	Median (Mean)	IQR (sd)	Min	Max	Median (Mean)	IQR (sd)	Min	Max	Median (Mean)	IQR (sd)	Min	Max
Population Density *	(5.649)	(1.461)	2.907	9.076	(5.885)	(1.352)	2.907	9.076	(5.196)	(1.234)	2.907	8.976
Accessibility *	(0.728)	(1.426)	−2.676	4.605	(0.855)	(1.353)	−2.676	4.605	(0.239)	(1.263)	−2.676	3.519
Public Facilities	3	1	1	5	3	1	1	5	3	1	1	5
Surrounding Environment	3	1	1	5	3	1	1	5	3	1	1	5
Public Safety	3	1	1	5	3	1	1	5	3	1	1	5
Emotional Attachment	4	1	1	5	4	1	1	5	4	2	1	5
Happiness	(7.534)	(2.269)	0	10	(7.484)	(2.250)	0	10	(6.904)	(2.403)	0	10
Life Satisfaction	(3.782)	(1.051)	1	5	(3.874)	(1.009)	1	5	(3.823)	(1.021)	1	5
Confidence in the Future	(4.053)	(1.046)	1	5	(4.144)	(1.004)	1	5	(4.031)	(1.052)	1	5
Income Satisfaction	3	2	1	5	3	1	1	5	3	1	1	5
Working Environment Satisfaction	3	1	1	5	4	1	1	5	3	1	1	5
Overall Job Satisfaction	3	1	1	5	3	1	1	5	3	1	1	5
Severity of Environmental Problems	(6.541)	(2.845)	0	10	(6.447)	(2.770)	0	10	(6.078)	(2.698)	0	10
Trust in Neighborhood	(6.895)	(2.260)	0	10	(6.894)	(2.218)	0	10	(6.455)	(2.258)	0	10
Net Income Per Capita (CNY) *	(8.517)	(0.798)	6.685	9.903	(8.181)	(0.647)	6.685	9.903	(7.976)	(0.691)	6.685	9.107
Biodiversity *	(6.978)	(5.765)	0	14.57	(4.453)	(5.531)	0	14.57	(6.925)	(5.853)	0	14.57

* We took the natural logarithm of these variables. Biodiversity = proportion of forest (and/or land with fruit trees) land area in administrative area. sd = standard deviation; IQR = interquartile range.

**Table 5 ijerph-20-04913-t005:** Total, direct and indirect effects of PBE on P-RCR in terms of Soc, Eco and Env.

Pathways and Effects	Dimensions	Model 1		Model 2		Model 3		Model 4	
		Point Estimate	Standard Error	Point Estimate	Standard Error	Point Estimate	Standard Error	Point Estimate	Standard Error
Eastern Regions									
PBE→PA→ Indirect Effects	Soc	**0.058** ***	0.010	**0.057** ***	0.010	**0.164** ***	0.029	**0.163** ***	0.029
	Eco	**0.031** ***	0.009	**0.031** ***	0.009	**−0.022** **	0.009	**−0.022** **	0.009
	Env	0.047	0.033	0.048	0.033	−0.067	0.064	−0.076	0.064
Direct Effects	Soc	0.172	0.031	0.172	0.031	0.274	0.091	0.277	0.091
	Eco	0.434	0.037	0.432	0.037	0.132	0.031	0.133	0.031
	Env	−0.654	0.122	−0.658	0.122	−0.711	0.230	−0.682	0.230
Total Effects	Soc	**0.230** ***	0.030	**0.230** ***	0.030	**0.438** **	0.085	**0.440** **	0.086
	Eco	**0.465** ***	0.035	**0.463** ***	0.035	**0.110** ***	0.029	**0.111** ***	0.029
	Env	**−0.607** ***	0.113	**−0.611** ***	0.113	**−0.778** **	0.213	**−0.758** **	0.214
Central Regions									
PBE→PA→ Indirect Effects	Soc	**0.021** **	0.008	**0.021** **	0.008	**0.102** ***	0.029	**0.101** ***	0.029
	Eco	0.005	0.009	0.005	0.009	0.001	0.008	0.000	0.007
	Env	**0.091** **	0.034	**0.091** **	0.034	−0.095	0.068	−0.075	0.068
Direct Effects	Soc	0.099	0.033	0.099	0.033	0.556	0.115	0.549	0.115
	Eco	0.494	0.038	0.492	0.038	0.090	0.029	0.089	0.028
	Env	−0.713	0.132	−0.712	0.132	0.666	0.262	0.765	0.271
Total Effects	Soc	**0.121** ***	0.031	**0.120** ***	0.031	**0.658** ***	0.111	**0.651** ***	0.111
	Eco	**0.499** ***	0.036	**0.497** ***	0.036	**0.091** **	0.027	**0.089** **	0.027
	Env	**−0.622** ***	0.123	**−0.622** ***	0.123	**0.571** **	0.247	**0.689** **	0.256
Western Regions									
PBE→PA→ Indirect Effects	Soc	0.013	0.008	0.014	0.008	0.040	0.029	0.040	0.029
	Eco	0.016	0.009	0.016	0.009	0.003	0.008	0.002	0.008
	Env	0.043	0.032	0.043	0.033	0.054	0.073	0.045	0.073
Direct Effects	Soc	0.081	0.031	0.086	0.031	0.226	0.116	0.239	0.116
	Eco	0.533	0.051	0.537	0.051	−0.090	0.035	−0.080	0.036
	Env	−0.461	0.142	−0.466	0.143	−0.150	0.299	−0.189	0.301
Total Effects	Soc	**0.094** **	0.030	**0.099** **	0.030	**0.267** *	0.110	**0.279** *	0.110
	Eco	**0.549** ***	0.048	**0.553** ***	0.049	**−0.087** **	0.033	**−0.078** *	0.033
	Env	**−0.418** **	0.135	**−0.423** **	0.135	−0.097	0.281	−0.144	0.283

Underlined and bold values represent significant total and mediation effects (5000 bootstrap samples, 95% bias-correct confidence level). *** *p* < 0.001; ** *p* < 0.01; * *p* < 0.05. PBE = perceived built environment; PA = place attachment; Soc = social dimension; Eco = economic dimension; Env = environmental dimension; “PBE→PA→” refers to the pathways from PBE to Soc/Eco/Env through PA.

**Table 6 ijerph-20-04913-t006:** Total, direct and indirect effects of OBE on P-RCR (eastern regions).

Pathways and Effects (Eastern Regions)	Dimensions	Model 1		Model 2		Model 3		Model 4	
		Point Estimate	Standard Error	Point Estimate	Standard Error	Point Estimate	Standard Error	Point Estimate	Standard Error
OBE→PBE→PA→ Indirect Effects	Soc	**−0.002** ***	0.001	**−0.002** ***	0.001	−0.005	0.002	−0.005	0.002
	Eco	−0.001	0.000	−0.001	0.000	0.001	0.000	0.001	0.000
	Env	−0.002	0.001	−0.002	0.001	0.002	0.002	0.002	0.002
OBE→PBE→ Indirect Effects	Soc	**−0.006** ***	0.002	**−0.006** ***	0.002	−0.009	0.004	−0.009	0.004
	Eco	−0.015	0.004	**−0.014** ***	0.004	−0.004	0.002	−0.004	0.002
	Env	0.022	0.007	0.022	0.007	**0.024** **	0.011	**0.022** **	0.010
OBE→PA→ Indirect Effects	Soc	**0.005** ***	0.002	**0.004** **	0.002	0.014	0.005	0.011	0.004
	Eco	0.003	0.001	**0.002** **	0.001	−0.002	0.001	−0.001	0.001
	Env	0.004	0.003	0.003	0.003	−0.006	0.006	−0.005	0.005
Total Indirect Effects	Soc	−0.003	0.003	−0.004	0.003	−0.001	0.006	−0.004	0.006
	Eco	−0.013	0.005	**−0.013** **	0.005	−0.006	0.002	−0.005	0.002
	Env	0.025	0.008	0.023	0.008	0.020	0.012	**0.020** *	0.012
Direct Effects	Soc	−0.018	0.009	−0.020	0.009	−0.013	0.028	0.000	0.029
	Eco	−0.003	0.010	−0.014	0.010	0.019	0.012	0.027	0.012
	Env	0.027	0.036	0.012	0.036	−0.314	0.076	−0.225	0.078
Total Effects	Soc	**−0.021** *	0.010	**−0.023** *	0.010	−0.014	0.028	−0.004	0.029
	Eco	−0.017	0.010	**−0.028** **	0.010	0.013	0.012	0.022	0.012
	Env	0.051	0.035	0.035	0.036	**−0.294** **	0.076	**−0.206** *	0.077

Underlined and bold values represent significant total and mediation effects (5000 bootstrap samples, 95% bias-correct confidence level). *** *p* < 0.001; ** *p* < 0.01; * *p* < 0.05. PBE = perceived built environment; PA = place attachment; Soc = social dimension; Eco = economic dimension; Env = environmental dimension. “OBE→PBE→PA→” refers to the pathways from OBE to Soc/Eco/Env first through PBE, then PA; “OBE→PBE→” refers to the pathways from OBE to Soc/Eco/Env through PBE. “OBE→PA→” refers to the pathways from OBE to Soc/Eco/Env first through PA.

**Table 7 ijerph-20-04913-t007:** Total, direct and indirect effects of OBE on P-RCR (central regions).

Pathways and Effects (Central Regions)	Dimensions	Model 1		Model 2		Model 3		Model 4	
		Point Estimate	Standard Error	Point Estimate	Standard Error	Point Estimate	Standard Error	Point Estimate	Standard Error
OBE→PBE→PA→ Indirect Effects	Soc	0.000	0.000	−0.001	0.000	−0.002	0.001	−0.003	0.001
	Eco	0.000	0.000	0.000	0.000	0.000	0.000	0.000	0.000
	Env	−0.002	0.001	−0.002	0.001	0.002	0.002	0.002	0.002
OBE→PBE→ Indirect Effects	Soc	−0.002	0.002	−0.002	0.002	−0.014	0.007	−0.014	0.007
	Eco	−0.011	0.006	−0.012	0.006	−0.002	0.001	−0.002	0.001
	Env	0.017	0.010	0.018	0.009	**−0.016** *	0.011	**−0.019** *	0.012
OBE→PA→ Indirect Effects	Soc	−0.001	0.001	−0.001	0.001	−0.005	0.004	−0.006	0.004
	Eco	0.000	0.001	0.000	0.001	0.000	0.000	0.000	0.000
	Env	−0.005	0.003	−0.005	0.004	0.004	0.005	0.004	0.005
Total Indirect Effects	Soc	−0.004	0.002	−0.004	0.002	−0.021	0.009	−0.022	0.009
	Eco	−0.012	0.006	**−0.013** *	0.006	−0.002	0.001	−0.002	0.001
	Env	0.010	0.009	0.010	0.009	−0.010	0.011	−0.013	0.012
Direct Effects	Soc	−0.003	0.010	−0.008	0.010	0.028	0.035	−0.008	0.035
	Eco	0.049	0.011	0.046	0.011	−0.077	0.010	−0.101	0.009
	Env	−0.045	0.044	−0.043	0.043	−0.453	0.083	0.324	0.085
Total Effects	Soc	−0.007	0.010	−0.013	0.010	0.007	0.035	−0.030	0.035
	Eco	**0.037** **	0.012	**0.033** **	0.012	**−0.079** ***	0.010	**−0.103** ***	0.009
	Env	−0.035	0.044	−0.033	0.043	**−0.463** ***	0.083	**0.311** ***	0.085

Underlined and bold values represent significant total and mediation effects (5000 bootstrap samples, 95% bias-correct confidence level). *** *p* < 0.001; ** *p* < 0.01; * *p* < 0.05. PBE = perceived built environment; PA = place attachment; Soc = social dimension; Eco = economic dimension; Env = environmental dimension. “OBE→PBE →PA→” refers to the pathways from OBE to Soc/Eco/Env first through PBE, then PA; “OBE→PBE→” refers to the pathways from OBE to Soc/Eco/Env through PBE. “OBE→PA→” refers to the pathways from OBE to Soc/Eco/Env first through PA.

**Table 8 ijerph-20-04913-t008:** Total, direct and indirect effects of OBE on P-RCR (western regions).

Pathways and Effects (Western Regions)	Dimensions	Model 1		Model 2		Model 3		Model 4	
		Point Estimate	Standard Error	Point Estimate	Standard Error	Point Estimate	Standard Error	Point Estimate	Standard Error
OBE→PBE→PA→ Indirect Effects	Soc	−0.001	0.000	−0.001	0.000	−0.002	0.001	−0.002	0.001
	Eco	−0.001	0.000	−0.001	0.000	0.000	0.000	0.000	0.000
	Env	−0.002	0.002	−0.002	0.002	−0.002	0.003	−0.002	0.004
OBE→PBE→ Indirect Effects	Soc	−0.004	0.002	−0.004	0.002	−0.010	0.006	−0.011	0.006
	Eco	**−0.024** ***	0.006	**−0.026** ***	0.006	**0.004** **	0.002	**0.004** **	0.002
	Env	0.021	0.008	0.023	0.008	0.007	0.013	0.009	0.015
OBE→PA→ Indirect Effects	Soc	−0.001	0.001	0.000	0.000	−0.002	0.002	−0.001	0.001
	Eco	−0.001	0.001	0.000	0.001	0.000	0.000	0.000	0.000
	Env	−0.002	0.002	−0.001	0.002	−0.002	0.004	−0.001	0.003
Total Indirect Effects	Soc	−0.005	0.002	−0.005	0.002	−0.013	0.006	−0.014	0.006
	Eco	**−0.025** ***	0.006	**−0.027** ***	0.006	**0.004** **	0.002	**0.004** *	0.002
	Env	0.017	0.008	0.019	0.008	0.002	0.013	0.006	0.014
Direct Effects	Soc	0.007	0.009	0.020	0.009	0.035	0.036	0.064	0.035
	Eco	−0.010	0.012	0.000	0.011	0.087	0.011	0.105	0.010
	Env	−0.019	0.043	−0.035	0.042	0.562	0.092	0.343	0.089
Total Effects	Soc	0.002	0.009	0.014	0.009	0.022	0.036	0.049	0.035
	Eco	**−0.035** **	0.012	**−0.028** **	0.011	**0.090** ***	0.011	**0.109** ***	0.009
	Env	−0.003	0.042	−0.016	0.041	**0.564** ***	0.090	**0.349** ***	0.088

Underlined and bold values represent significant total and mediation effects (5000 bootstrap samples, 95%. bias-correct confidence level). *** *p* < 0.001; ** *p* < 0.01; * *p* < 0.05. PBE = perceived built environment; PA = place attachment; Soc = social dimension; Eco = economic dimension; Env = environmental dimension. “OBE→PBE→PA→” refers to the pathways from OBE to Soc/Eco/Env first through PBE, then PA; “OBE→PBE→” refers to the pathways from OBE to Soc/Eco/Env through PBE. “OBE→PA→” refers to the pathways from OBE to Soc/Eco/Env first through PA.

## Data Availability

Publicly available datasets were analyzed in this study. These data can be found here: [http://www.isss.pku.edu.cn/cfps/ (accessed on 21 June 2022)].

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
