# Peer review of "The Proactive Effects of Built Environment on Rural Community Resilience: Evidence from China Family Panel Studies"

_ijerph, 2023, doi:10.3390/ijerph20064913_

Round 1
Reviewer 1 Report
The paper is straightforward enough about the rationale for inquiry, but I was nevertheless concerned on a couple of points:
1. These data are at this point very dated, given that it's 2022 as of this review. Interpretations will also be out of phase with current realities.
2. Study is not fully placed in context of extant research.
3. Not always clear how the specific study is making use of data/questions in source data - as far as indicators. A section should be added on these data and whatever is being used for this study, more fully aligning the relationship between research questions and data utilized.
4. The use of acronyms is a lot to deal with for a wide-ranging audience.
5. Why this statistical approach and not some other?
6. Some limitations are mentioned but they are not fully addressed.
7. Reasons are given for use of the China case but it could be argued that this is simply an argument for use of this (out of date) dataset.
8. Points being made and ultimate value for policy-making and use beyond this paper are not as clear as they should be.
Author Response
Point 1: These data are at this point very dated, given that it's 2022 as of this review. Interpretations will also be out of phase with current realities.
Response 1: We thank the reviewer for raising this point. We added discussions of these concerns into the limitation section of the revised manuscript. Specifically, the following changes have been made ( see PDF version, page16, line 568-578):
“At last, the datasets we use are not recent, though these are the most recent datasets available in relation to BE in the released CFPS datasets. However, we believe that the advantages of using these datasets are evident and our findings are still valid for the current realities. The reasons are as follows. First, these datasets are longitudinal and nationally representative, providing high-quality and large sample size data. Moreover, they can be updated in the future and facilitate our follow-up study with panel data. Second, at the core of our study is the BE-P-RCR relationship that closely relates to psychological factors not easily changing with time, such as human cognitions, emotions and behaviors. This means that for this study, time factor might not be decisive. The consistency between our findings and some earlier study results discussed in the discussion section might be seen as evidence.”
Point 2: Study is not fully placed in context of extant research.
Response 2: We thank the reviewer for the comment. In response to this comment, the following changes have been made in the revised manuscript(see PDF version, page 2, line 86-88):
“However, although a few researchers test the links between rural people’s perceptions of facilities and P-RCR in case studies[2], the implications of OBE and PBE have not yet been further examined simultaneously and holistically in empirical studies on P-RCR..”
Point 3: Not always clear how the specific study is making use of data/questions in source data - as far as indicators. A section should be added on these data and whatever is being used for this study, more fully aligning the relationship between research questions and data utilized.
Response 3: We thank the reviewer for the comment. We add section 3.2.5 in the revised manuscript to explain the specific questions used in this study. Specifically, the following changes have been made(see PDF version, page 8, line 327-333):
3.2.5. Questions used for variable measurement
Specific questions used to derive indicators for PBE, PA, different dimensions of P-RCR and the covariate are displayed in Table 3.
Table 3. Questions used to derive indicators for PBE, PA, P-RCR and the covariate.
|
Variables |
Indicators |
Questions |
Source |
|
|
PBE |
|
Community Environment |
How is the surrounding environment of your community (noise, trash disposal etc.)? (reversed code ranging from 1= very poor to 5= very good) |
CFPS 2016 Full Questionnaires |
|
Safety |
How is the public safety around your community? (reversed code ranging from 1= very poor to 5= very good) |
|||
|
Public Facilities |
How do you think of the public facilities around your community? (reversed code ranging from 1= very poor to 5= very good) |
|||
|
P-RCR |
Individual-level |
Social Dimension |
1. Are you happy? (ranging from 1=lowest to 10=highest) 2. How confident are you about your future? (ranging from 1=not confident at all to 5=very confident) 3. Are you satisfied with your life? (ranging from 1=very unsatisfied to 5=very satisfied) |
CFPS 2014 Full Questionnaires; CFPS 2016 Full Questionnaires |
|
Economic Dimension |
1. How satisfied are you with your current income from this job? (ranging from 1=very unsatisfied to 5=very satisfied) 2. In general, how satisfied are you with this job? (ranging from 1=very unsatisfied to 5=very satisfied) 3. How satisfied are you with the working environment in this job? (ranging from 1=very unsatisfied to 5=very satisfied) |
|||
|
Environmental Dimension |
How would you rate the severity of the environmental problem in China? (ranging from 1=not severe to 10=extremely severe) |
|||
|
Community-level |
Social Dimension |
How much do you trust your neighborhood? (ranging from 1=distrustful to 10=very trustworthy) |
CFPS 2014 Full Questionnaires |
|
|
Economic Dimension |
The net income per capita in your village (yuan) |
|||
|
Environmental Dimension |
1.The total area of forest and/or land with fruit trees in your village (mu) 2. What is the current administrative area of your village/residential community?( kilometer2/mu) |
|||
|
PA |
|
Emotional Attachment |
Are you emotionally attached to your community? (ranging from 1=very good to 5=very poor) |
CFPS 2016 Full Questionnaires |
|
Covariate |
|
Self-reported Socioeconomic Status |
1. What is your relative income level in your local area? (ranging from 1=lowest to 5=highest) 2. What is your social status in your local area? (ranging from 1=lowest to 5=highest) |
CFPS 2014 Full Questionnaires |
BE=built environment; PBE=perceived built environment; P-RCR= the proactive aspect of rural community resilience; PA=place attachment. CFPS 2014/2016= China family Panel Studies in 2014/2016.
Point 4: The use of acronyms is a lot to deal with for a wide-ranging audience.
Response 4: : We would like to thank the reviewer for pointing out this issue. In order to minimize readers’ inconveniences with acronyms, we replace and explain some acronyms with their full terms in the revised manuscript. Specifically, the following changes have been made:
- Mdn is replaced with median(see PDF version, page 10, line 389-397);
- The meaning of sd and IQR are explained in the text ( see PDF version, page 10, line 385, line 390);
- CMIN/DF is replaced with Chi-square/degrees of freedom (see PDF version, page 11, line 408-409)ï¼›
- CFI is replaced with comparative fit index(see PDF version, page 11, line 409)ï¼›
- RMSEA is replaced with root mean square error of approximation(see PDF version, page 11, line 409-410)ï¼›
- SRMR is replaced with standardized root mean square residual(see PDF version, page 11, line 410);
- E. is replaced with Standard Error in Table (see PDF version, page 11, line 436; page 12, line 458; page 13, line 465; page 13, line 472).
Point 5: Why this statistical approach and not some other?
Response 5: We thank the reviewer for the comment. We apologise that our manuscript might have been unclear in stating the reasons why we use Structural Equation Modeling as our statistical approach not others. In order to clarify this statement, the following reasons have been added in the methods section of the revised manuscript (see PDF version, page 9, line 339-347):
“The reasons we use SEM as an analytical tool in this study are manifold. First, SEM allows researchers to investigate complex relationships between multiple constructs in a single model and provides an easier way to discuss the model[55, 69].Therefore, it fits well with our study that attempt to explore the associations between OBE, PBE, PA and three different dimensions of P-RCR in one holistic framework. Second, SEM is usually applied to verify theoretical hypothesis by analyzing observation and latent variables through statistical procedures including path analysis, regression and structural equations [55, 57]. As a result, it can be a useful tool for testing the BE-P-RCR relationship we postulate in this study.”
Point 6: Some limitations are mentioned but they are not fully addressed.
Response 6: We thank the reviewer for the comment. We discussed the advantages of using CFPS datasets in the limitation section of the revised manuscript to address our data related limitations. Specifically, the following changes have been made (see PDF version, page 16, line 568-573):
“At last, the datasets we use are not recent, though these are the most recent datasets available in relation to BE in the released CFPS datasets. However, we believe that the advantages of using these datasets are evident and our findings are still valid for the current realities. The reasons are as follows. First, these datasets are longitudinal and nationally representative, providing high-quality and large sample size data. Moreover, they can be updated in the future and facilitate our follow-up study with panel data.”
Point 7: Reasons are given for use of the China case but it could be argued that this is simply an argument for use of this (out of date) dataset.
Response 7: We appreciate the reviewer’s feedback. We apologise that our manuscript might have been unclear in stating the reasons for the use of the China case and CFPS datasets. In order to clarify these statements, the following changes have been made in the revised manuscript:
- Another reason for using China case has been added in the introduction section (see PDF version, page 2, line 95-97):
“First, compared to developed countries, resilience research on communities attracts limited attention in developing economies with a particular gap in research in China regarding P-RCR [33].”
- The reasons for use of CFPS datasets have been added in the limitations (see PDF version, page16, line 568-578)
“At last, the datasets we use are not recent, though these are the most recent datasets available in relation to BE in the released CFPS datasets. However, we believe that the advantages of using these datasets are evident and our findings are still valid for the current realities. The reasons are as follows. First, these datasets are longitudinal and nationally representative, providing high-quality and large sample size data. Moreover, they can be updated in the future and facilitate our follow-up study with panel data. Second, at the core of our study is the BE-P-RCR relationship that closely relates to psychological factors not easily changing with time, such as human cognitions, emotions and behaviors. This means that for this study, time factor might not be decisive. The consistency between our findings and some earlier study results discussed in the discussion section might be seen as evidence.”
We hope these changes make the explanations for using Chinese rural communities and the CFPS datasets clearer.
Point 8: Points being made and ultimate value for policy-making and use beyond this paper are not as clear as they should be.
Response 8: We thank the reviewer for the comment. Based on our findings, we added three recommendations for rural community development and P-RCR enhancement in China or other developing countries in the revised manuscript. Specifically, the following changes have been made(see PDF version, page 16, line 591-603):
“Based on our findings, the following recommendations are offered for rural com-munity development and P-RCR enhancement in China or other countries or regions facing similar rural issues as China:
- Improvements to the rural built environment, such as new rural reconstruction and rural settlement remediation, should not only focus on infrastructure constructions while ignoring people’s perceptions and evaluations of their surrounding environment.
- Top-down planning activities initiated by the government should develop more detailed and targeted planning schemes for rural service accessibility and village merger, which will be helpful for increasing P-RCR in different regions.
- The development and implementation of built environment policies should con-sider rural people's emotion ties with their communities, including both the pros and cons of these emotion ties for P-RCR.”

Reviewer 2 Report
Nice work. Minor edits.

Author Response
We thank the reviewer for the comment. As the reviewer suggested, the following changes have been made in the revised manuscript (see PDF version):
Line 38, paragraph 1. “of” has been added after “state”. line 42, paragraph 1. “contains” has been replaced with “combines”.
Line 84, 85, page 2. Unclear layout have been corrected.
Line 102, page 3. The sentence has been rewritten as “Second, what those resilience communities needed to improve urgently is the proactive aspect”.
Line 103, page 3. “the” has been added before “rural population”; Line 104, “in” has been added before “1995”.
Line 104-105, page 3. “the” has been added before “rural population”, “reduced about” has been replaced with “declined by”.
Line106, page 3. “young people” has been replaced by “rural labourers”. “is the” has been added before “reduction”.
Line 109, page 3. “subjects” has been replaced with “cohesion”.
Line 160, page 4. “person” has been replaced with “people”.
Line 170, page 4. “Soc, Eco and Env” has been replaced with “Social, economic and environmental”.

Reviewer 3 Report
I want to congratulate and thank the authors because it has been a long time since I read such an interesting article and so well written and structured.
However, I think that, given the importance of the theme of the article, it should be accessible to readers of different specialties, so I recommend that some acronyms be defined (see comments).
Continuation of the good work!
· The authors state that the research relied on 7528 respondents from rural areas across most of China territory, but they never say how representative this sample is and whether a qualitative or quantitative analysis is intended.
· Line 96, paragraph 1. Should be “Second, what those resilience communities needed to improve urgently is the proactive aspect” or “Second, those resilience communities needed to improve urgently its proactive aspect” and not “Second, those resilience communities needed to im-96 prove urgently is the proactive aspect.” Please find a correct way to write this sentence.
· In chapter 2 nothing is said about the OBE-P-RCR direct link. How does OBE directly influence P-RCR?
· Line 140, paragraph 2. In the model presented in figure 1, “Theoretical framework of the BE-P-RCR relationship”, the P-RCR only receives influence, it does not influence any of the other characteristics, but, in fact, shouldn't this model be more “dynamic”? After all, the P-RCR will influence any of the other three factors.
· Line 141, paragraph 2. Figure 1 and its caption are apart.
· Line 222, paragraph 3. This phrase makes no sense: it begins "Since the variables in our study involved many aspects of rural life that connect with several CFPS datasets released so far… " but then does not conclude.
· Line 222, paragraph 3. Table 1 could come immediately after line 222, where this table is referred.
· Line 274, paragraph 3.2.3. Title 3.2.3. and its text are apart.
· Line 282 and 283, paragraph 3.2.3. There’s a comma out of place, please verify.
· Line 321, 325 and 3333, paragraph 3.3. The enumeration of the three points is outside the text limits, please format this correctly.
· The figures have no explicit authorship in the caption. Please check this with the publication rules
· Line 358, paragraph 4.1. What does sd means? Its meaning is not explained in the text. You do not have to explain the method, but you must explain what the acronyms mean.
· Line 361, paragraph 4.1. What does IQR and Mdn means? Its meaning is not explained in the text.
· Line 377, paragraph 4.2. What does CMIN/DF, CFI, RMSEA and SRMR means? Its meaning is not explained in the text.
· Line 392, paragraph 4.2.1. Title 4.2.1. and its text are apart.
· Line 403, paragraph 4.2.1. Table 4. What does S.E. means? Its meaning is not explained.
· Line 408, paragraph 4.2.2. Title 4.2.2. and its text are apart.
· Line 432, paragraph 4.2.2. Table 6 is in two pages. Is it possible for this table to be all on one page?
· Line 574, Appendix. Table A2 is in two pages. Is it possible for this table to be all on one page?
Reading the article is complicated by the number of acronyms used, although most are explained. However, I do not think that the authors can present another option and the one they use seems to me to be the simplest, despite everything.
Author Response
Point 1: The authors state that the research relied on 7528 respondents from rural areas across most of China territory, but they never say how representative this sample is and whether a qualitative or quantitative analysis is intended.
Response 1: Thank you for the comment. We added the sentence “This sample is nationally representative and has been divided into eastern, central, and western region groups according to communities' geographic locations.” in introduction section of the revised manuscript (see PDF version, page 2, line 92-94).
Point 2: Line 96, paragraph 1. Should be “Second, what those resilience communities needed to improve urgently is the proactive aspect” or “Second, those resilience communities needed to improve urgently its proactive aspect” and not “Second, those resilience communities needed to im-96 prove urgently is the proactive aspect.” Please find a correct way to write this sentence.
Response 2: Thank you for the comment. This sentence has been rewritten as “Second, what those resilience communities needed to improve urgently is the proactive aspect” in the revised manuscript (see PDF version, page 3 , line 102-103).
Point 3: In chapter 2 nothing is said about the OBE-P-RCR direct link. How does OBE directly influence P-RCR?
Response 3: We thank the reviewer for raising this point. We apologise that our manuscript might have been unclear in stating the direct influences of OBE on P-RCR. In order to clarify this statement, the following changes have been made in the revised manuscript:
- We revised the caption of 2.2 (see PDF version, page 4, line 168 )
“2.2 Direct Paths From OBE/PBE to P-RCR with three fundamental dimensions”
- We added captions of 2.2.1 and 2.2.2 (see PDF version, page 4, line 169, line 187)
“2.2.1. Three fundamental dimensions of P-RCR”
“2.2.2. The influences of OBE/PBE on P-RCR”
- We added section 2.2.3. (see PDF version, page 5, line 206-209)
“2.2.3. Direct paths from OBE/PBE to different dimensions of P-RCR
It should be noted that the abovementioned influences of OBE/PBE on P-RCR might be direct, indirect or both. We assume that OBE/PBE affects P-RCR in both ways. Direct paths from OBE/PBE to P-RCR are included in the framework.”
We hope that it is clearer now.
Point 4: Line 140, paragraph 2. In the model presented in figure 1, “Theoretical framework of the BE-P-RCR relationship”, the P-RCR only receives influence, it does not influence any of the other characteristics, but, in fact, shouldn't this model be more “dynamic”? After all, the P-RCR will influence any of the other three factors.
Response 4: We thank the reviewer for raising this point. In response to the reviewers’ comments, we added the discussion of the potential impacts of P-RCR on BE into limitation section of the revised manuscript. Specifically, the following changes have been made (see PDF version, page 15, line 562-568):
“We can not verify the potential reciprocal causation between BE and P-RCR in this study, though resilient rural communities may intentionally increase built capital investments that improve BE qualities. This is not only because the exact pathways and scales by which P-RCR affects BE are still unclear, but also the cross-sectional data can not statistically estimate reciprocal causation due to the lack of temporal precedence[73]. An improved model and panel data are needed in future studies.”
Point 5: Line 141, paragraph 2. Figure 1 and its caption are apart.
Response 5: Thank you for the comment. It has been modified in the revised manuscript (see the PDF version, page 4, line 149).
Point 6: Line 222, paragraph 3. This phrase makes no sense: it begins "Since the variables in our study involved many aspects of rural life that connect with several CFPS datasets released so far… " but then does not conclude.
Response 6: We thank the reviewer for the comment. We apologise for the confusion generated by this phrase. We rewrite the sentence in the revised manuscript (see PDF version, page 5, line 236-238). Specifically, the following changes have been made:
“We used different datasets and waves of CFPS (Table 1), because the variables in our study involved many aspects of rural life that connect with several CFPS datasets released so far, and parts of these variables were collected in different waves.”
We hope that it is clearer now.
Point 7: Line 222, paragraph 3. Table 1 could come immediately after line 222, where this table is referred.
Response 7: Thank you for the comment. Table 1 has been inserted into the text immediately after its first citation in the revised manuscript (see PDF version, page 6, line 241-243).
Point 8: Line 274, paragraph 3.2.3. Title 3.2.3. and its text are apart.
Response 8: Thank you for the comment. It has been modified in the revised manuscript (see PDF version, page 7, line 290).
Point 9: Line 282 and 283, paragraph 3.2.3. There’s a comma out of place, please verify.
Response 9: Thank you for the comment. This has been corrected in the revised manuscript (see PDF version, page 7, line 298).
Point 10: Line 321, 325 and 3333, paragraph 3.3. The enumeration of the three points is outside the text limits, please format this correctly.
Response 10: Thank you for the comment. This has been corrected in the revised manuscript (see PDF version, page 9, line 348, line 352, line 360).
Point 11: The figures have no explicit authorship in the caption. Please check this with the publication rules.
Response 11: Thank you for the comment. “Source: authors” has been added in the caption in the revised manuscript (see PDF version, page 9, line 359).
Point 12: Line 358, paragraph 4.1. What does sd means? Its meaning is not explained in the text. You do not have to explain the method, but you must explain what the acronyms mean.
Response 12: Thank you for the comment. We replaced “sd” with its full term “standard deviation“ where “sd” is used for the first time. Specifically, the following changes have been made in the revised manuscript(see PDF version, page 10 ,line 384-385):
“Table 4 displays that the average population density (natural logarithm) and the standard deviation(sd) in eastern, central and western regions are 5.649 (sd=1.461), 5.885 (sd=1.352), 5.196 (sd=1.234) respectively.”
Point 13: Line 361, paragraph 4.1. What does IQR and Mdn means? Its meaning is not explained in the text.
Response 13: Thank you for the comment. We replaced “IQR” with its full term “interquartile range “ where “IQR” is used for the first time and replaced “Mdn” with “Median” in the text. Specifically, the following changes have been made in the revised manuscript(see PDF version, page 10, line 388-391):
“Most respondents evaluate PBE as fair, for the median values of public facilities, sur-rounding environment and public safety in three regions are all 3 (“fair” option), while the interquartile range (IQR) of these values are all 1, which represents a central tendency of these values.”
Point 14: Line 377, paragraph 4.2. What does CMIN/DF, CFI, RMSEA and SRMR means? Its meaning is not explained in the text.
Response 14: We thank the reviewer for the comment. In the revised manuscript(see PDF version, page 11, line 408-410), we replaced CMIN/DF with Chi-square/degrees of freedom, CFI with comparative Fit Index, RMSEA with root mean square error of approximation and SRMR with standardized root mean square residual.
Point 15: Line 392, paragraph 4.2.1. Title 4.2.1. and its text are apart.
Response 15: Thank you for the comment. It has been modified in the revised manuscript (see PDF version, page 11, line 425).
Point 16: Line 403, paragraph 4.2.1. Table 4. What does S.E. means? Its meaning is not explained.
Response 16: Thank you for the comment. We replaced “S.E.” with “Standard Error” in Table 5, Table 6, Table 7 and Table 8 in the revised manuscript(see PDF version, page 11, line 436; page 12, line 458; page 13, line 465).
Point 17: Line 408, paragraph 4.2.2. Title 4.2.2. and its text are apart.
Response 17: Thank you for the comment. It has been modified in the revised manuscript (see PDF version, page 12, line 441).
Point 18: Line 432, paragraph 4.2.2. Table 6 is in two pages. Is it possible for this table to be all on one page?
Response 18: Thank you for the comment. It has been modified in the revised manuscript ( see PDF version, page 13, line 465).
Point 19: Line 574, Appendix. Table A2 is in two pages. Is it possible for this table to be all on one page?
Response 19: Thank you for the comment. It has been modified in the revised manuscript ( see PDF version, page 17, line 635).

Round 2
Reviewer 1 Report
The authors have responded acceptably to my previous comments, though I continue to think the value of the research contribution is ultimately marginal at best.